# Deformation Control of the Existing Medium-Low-Speed Maglev Metro Viaduct over a Double-Line Bored Tunnel

**Hua Peng \*** , **Zikai Wu, Xv Peng, Xiaoqi Xiao and Zichen Li**

School of Civil Engineering, Beijing Jiaotong University, Beijing 100044, China
* Correspondence: hpeng@bjtu.edu.cn

**Abstract:** Taking a double-line underground excavation subway tunnel underneath the medium-low-speed Maglev bridge section of the Beijing S1 Metro line as a project reference, the deformation control index of the Maglev metro bridge and track panel was studied using numerical simulation and data monitoring, under the conditions of deep-hole grouting measures for the soil layer around the excavation tunnel and active lifting for the superstructure of the viaduct. The results of the numerical simulation show that the maximum deformation value of the pier is −1.73 mm and the maximum vertical deformation value of the Maglev rail track is −1.56 mm. Monitoring shows that the maximum vertical deformation value of the magnetic levitation rail is −1.25 mm and the deformation is within the allowable range. Both numerical simulation and field monitoring results reflect the deformation trend of existing structures; the deep-hole grouting measures taken to strengthen the soil layer around the tunnel and the upper structure of the active roof elevation frame can effectively control the deformation of the existing structure. The study can provide guidance and reference for similar projects.

**Keywords:** double-line subway-bored tunnel; Maglev metro; under-traversed bridge; deformation control index

## 1. Introduction

In recent years, more and more Maglev tracks have been used in subway lines in major cities. Compared with traditional rail transit, Maglev tracks have many advantages, such as safety, reliability, low vibration, low energy consumption, and small turning radius. Maglev lines are mostly built in the form of a viaduct. Maglev tracks have special requirements for deformation, and the allowable value of deformation in crossing projects is strict. Therefore, if the tunnel passing underneath causes certain deformation of the existing viaduct bridge structure, it will seriously threaten the operational safety of the Maglev track.

To date, many scholars have conducted research on deformation control of urban rail transit bridges under tunnels [1–7]. Li Song et al. [8] analyzed the influence of a shield tunnel underpass on the deformation of an adjacent viaduct pile foundation. Cui Guangyao et al. [9] found that the setting of isolation piles can effectively control the deformation of high-speed railway viaducts under the influence of tunnels crossing underneath. Wang Guofu et al. [10] studied the deformation control effect of three active pre-support technologies, namely, the frame structure, triaxial mixing pile structure, and separation wall structure, using a numerical calculation method based on the project of a subway shield tunnel passing through an existing viaduct at a close distance in Jinan. The medium-and low-speed Maglev system has very strict requirements on the deformation of the track beam and the substructure [11]. For research on tunnels passing through Maglev viaducts, Liu Zhigang [12] studied the deformation control and construction safety problems of the shield tunnel passing through the Maglev viaduct in a soft soil layer by combining theoretical analysis, field testing, and actual data analysis. Zhang Xiaofeng et al. [13] analyzed the deformation monitoring data of a pipe jacking underpass in a Maglev road

viaduct project, and verified the feasibility of the method of controlling the deformation of existing structures through real-time monitoring and modifying construction parameters. Guo [14] studied the deformation control effects of three protective measures, pre-grouting reinforcement of pile foundation, active underpinning of bridge pile foundations, and pile foundation reinforcement, based on the simulation calculation results of a shield tunnel passing under an existing railway bridge in Shenzhen. Peng Hua et al. [15] analyzed the monitoring data and numerical simulation results of a large-diameter shield underpass viaduct project. Combined with engineering experience, they summarized the reasonable tunneling parameters and control technology in shield crossing. Li Haiyang [16] studied the settlement control problem of a four-line overlapping small spacing tunnel crossing under an old bridge, and put forward the river bed hardening measures of wrapped grouting reinforcement and jet grouting piles combined with concrete slab bottom protection. Han [17] studied the influence of three important factors on the deformation of pile foundation, namely, the buried depth of the tunnel, the relative position of the pile foundation to the horizontal axis of tunnel and the fracture surface, and the eccentricity ratio of the tunnel through the pile foundation, by combining numerical simulation and data monitoring.

Meanwhile, the soil–structure interaction is also a key point in scholars' research. Tullini, N et al. [18] studied the buckling deformation of finite-length Timoshenko beams in frictionless contact with an elastic substrate in a generalized plane stress or plane strain regime. Gao, Q et al. [19] studied the soil–structure interaction emanating from seismic stationary random excitations using the pseudo-excitation method in combination with the precise integration method. The result shows that the soil–structure interaction is very significant under the condition of a large structure mass or soft soil.

Although the above research has been undertaken, there is no research on deformation control of an underground tunnel passing through an existing Maglev viaduct metro line. This study took the magnetic suspension viaduct section project of a double-track subway tunnel crossing under the existing Beijing Line S1 as an example. Under the condition of deformation control measures using deep-hole grouting [20–24] to advance the reinforcement of the soil around the tunnel and actively jacking the superstructure of the bridge [25–29], the deformation of the existing structure's viaduct pier and the track panel of the magnetic suspension track, caused by the crossing of the double track subway tunnel underneath, was calculated numerically and verified by data measured in the field.

## 2. Background

### 2.1. Project Overview

A double-track subway tunnel passes under the existing Beijing S1 medium-low-speed Maglev line. The tunnel has a standard horseshoe section. The outer contour size of the interval structure is 6.48 m × 6.62 m, the initial lining thickness is 0.25 m, and the secondary lining thickness is 0.3 m. The new double-line subsurface excavation tunnel was excavated from north to south, and the step method was used for excavation. The tunnel curve section passes through the existing magnetic suspension elevated section at an angle of 80°, and the tunnel overburden is about 16.0 m. The flat and cross-section positions of the new double-line subsurface excavation tunnel and the existing line are shown in Figures 1 and 2.

The existing middle- and low-speed magnetic levitation elevated section is a double line, the structure foundations are piles, the pile length is 17 m, and the diameter is 1 m. The existing viaduct under the area of influence is a three-span continuous beam structure with a bridge span of 23.6 m + 39.2 m + 23.6 m. The new section crosses the middle span of the continuous beam bridge, and the crossing pier numbers are 72# and 73#. The minimum plane distance between the new right tunnel and the pier 72# bridge pile is about 5.9 m, and the minimum plane distance between the left tunnel and the pier 73# bridge pile is about 4.5 m. The bottom elevation of the new tunnel structure is about 1.0 m deeper than that of the existing bridge pile.

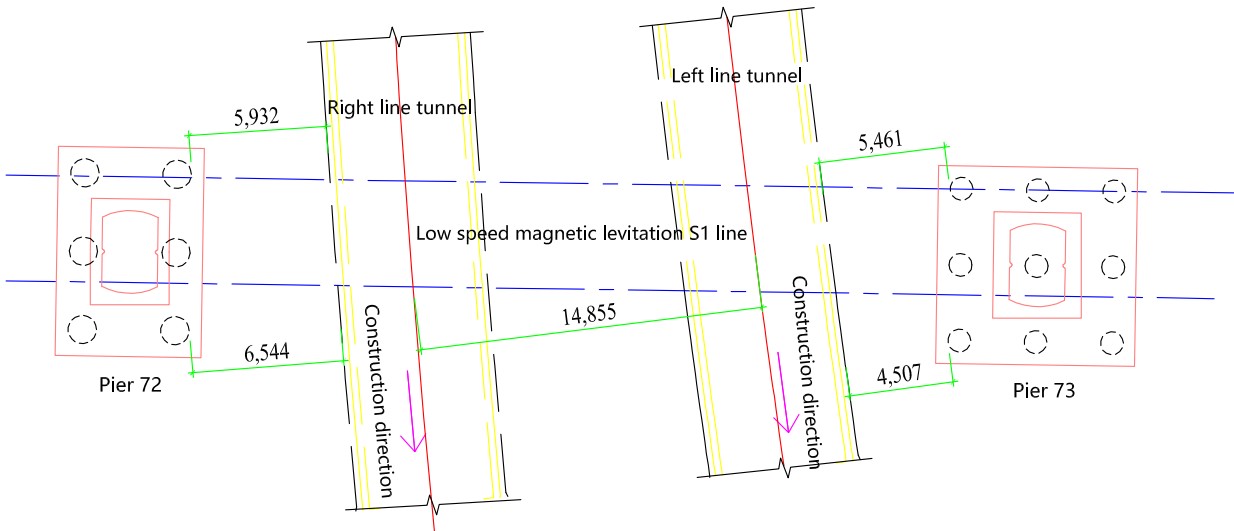

**Figure 1.** Plan of new subway excavated tunnel and existing viaduct.

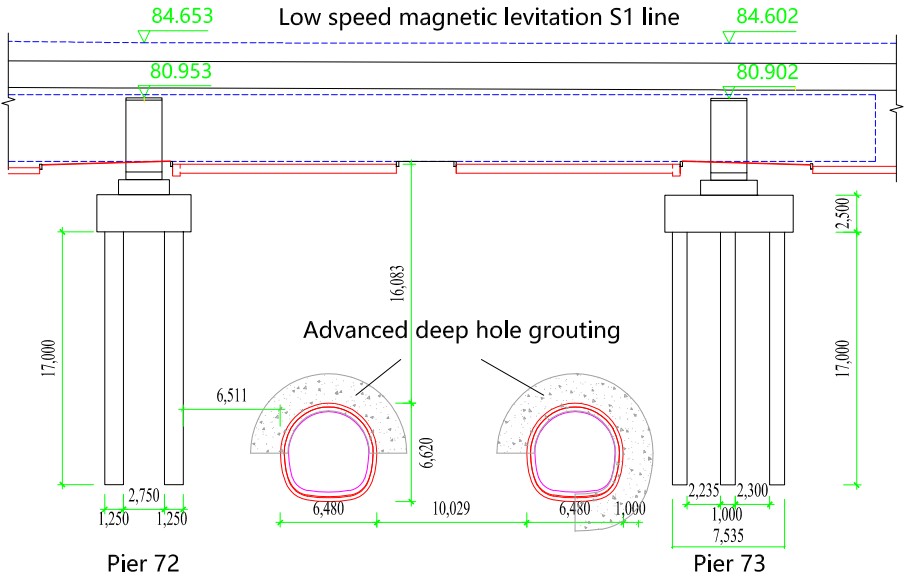

**Figure 2.** Elevation of new subway excavated tunnel and existing viaduct.

The maximum depth of the exposed stratum is 70 m. According to the drilling data and indoor geotechnical test results, and according to the sedimentary age and genetic type of the stratum, the soil layer within the exploration scope of the project site is divided into four layers: artificial accumulation layer (Qml), newly deposited alluvial-diluvial layer (Q42 + 3al + pl), Quaternary late Pleistocene alluvial-diluvial layer (Q3al + pl), and Quaternary late Pleistocene slope diluvial layer (Q3dl + pl). This section of the line tunnel through the surrounding rock mainly comprises a clay silt fill ① layer, miscellaneous fill ①$_1$ layer, silty clay ②$_1$ layer, pebble ⑤ layer, pebble ⑦ layer, sandy silt clay silt ⑦$_3$ layer, pebble ⑧$_1$ layer, silty clay ⑧ layer, and pebble ⑨ layer. The groundwater depth is 10.0~25.0 m. The stratigraphic profile is shown in Figure 3.

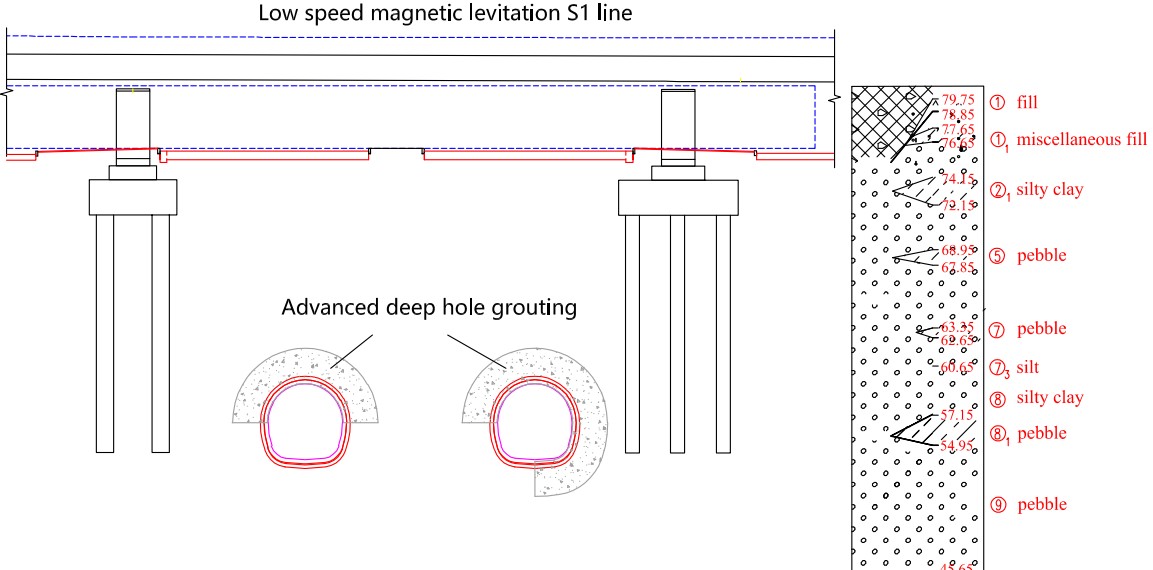

**Figure 3.** Elevation with geological section.

*2.2. Deformation Control Measures for Double-Track Excavated Tunnel Passing under a Magnetic Suspension Viaduct*

In order to control the influence of the tunnel passing underneath on the deformation of the existing viaduct and the Maglev track, it is necessary to take deformation control measures. In crossing engineering, deformation control measures are mainly divided into active control measures and passive control measures. The construction of the tunnel under the tunnel will have a certain impact on the existing structure, and the Maglev track structure has more stringent requirements for deformation control than the traditional wheel–rail line. Therefore, the project also adopts a combination of active and passive schemes:

(1) Deep-hole grouting is used in advance to reinforce the surrounding soil of the structure, and the cross-section is 2 m outside the initial support contour. The grouting slurry is a cement–water glass double slurry. The reinforced soil should have good uniformity and self-reliance, and the unconfined compressive strength of the reinforced soil should not be less than 0.8 MPa.

(2) During the construction of the interval crossing bridge piles, the step method with a temporary inverted arch is adopted. The initial lining structure controls the excavation step of 0.5 m (the grid spacing is 0.5 m), and the temporary inverted arch is closed in time. The left and right lines of the tunnel are staggered 15 m, and the upper and lower steps are staggered 3 m.

(3) Grouting behind the primary support is carried out in time during the construction of the primary support, and grouting behind the secondary lining is carried out in time during the construction of the secondary lining, so as to strictly control the grouting pressure and grouting amount and ensure the grouting effect.

(4) The lifting scheme for the bridge superstructure is taken as a safety plan. During the construction period, jacks are arranged at the top of the pier to implement synchronous lifting. The jack arrangement is shown in Figure 4. Real-time monitoring is carried out during tunnel construction. Once the settlement of the main beam reaches 2 mm, the main beam is jacked up with a jack during the empty window period of the S1 line at night, and a height-adjusting steel plate is added between the bottom of the beam and the support to ensure that the deformation of the main beam structure does not exceed the limit.

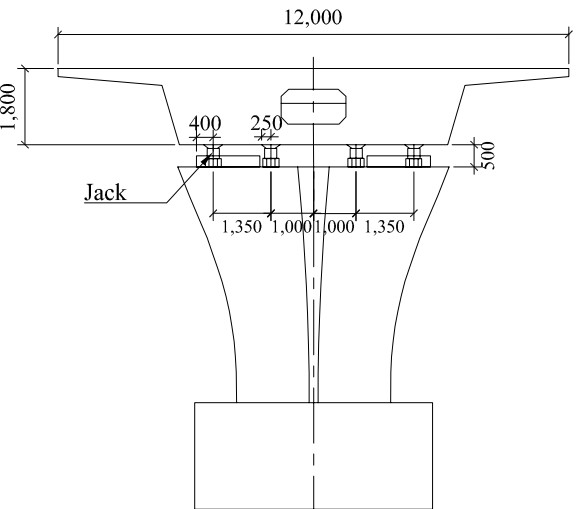

**Figure 4.** Layout of lifting jack arrangement.

## 2.3. Magnetic Levitation Rail Transit Control Standard

The affected section adopts the structure of a medium-low-speed Maglev track. The medium-low-speed Maglev traffic track is paved with rails as the unit, which has the function of supporting the Maglev vehicle and bearing the suspension force, guiding force, traction force, and braking force of the vehicle.

The rail platoon is the equipment used for the suspension, driving, and parking of the train. It guides the operation of the locomotive and directly bears the load from the vehicle and transmits it to structures such as bridges and piers. Therefore, the rail platoon must be strong and stable and have the correct geometry to ensure the safe operation of the vehicle. The rail row is mainly composed of an F-shaped steel guide rail (overlying the induction plate), rail sleeper, joint components, and connecting bolts. Under the rail row, the rail-bearing platform made of reinforced concrete piers is connected by fasteners with the rail-bearing platform. The layout of the track structure section is shown in Figures 5 and 6.

According to the 'Code for Design of Medium and Low Speed Maglev Traffic' (CJJ/T262-2017) 'Table 7.4.1 Static Smoothness of Low Speed Maglev Track' [30] and the requirements of Beijing metro operation units, the requirements for track smoothness are shown in Table 1.

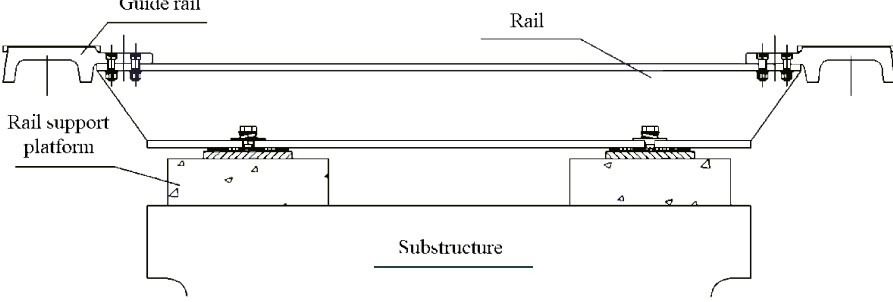

**Figure 5.** Cross-section of Maglev rail track.

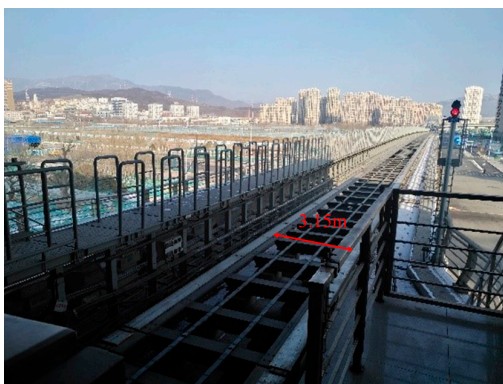

**Figure 6.** On-site photos of Maglev rail.

**Table 1.** Static displacement of medium- and low-speed Maglev track.

| Item | Threshold Value | Note |
|------|-----------------|------|
| Rail alignment deviation | ±3 mm | Center distance between two tracks; the value under the condition of track-locking temperature. |
| Horizontal level deviation | ±3 mm | / |
| Vertical height deviation | ±3 mm | Front and rear height; measuring chord length 10 m. |
| Twist | ±1 mm | Measuring chord length 4 m. |
| Coplanarity of magnetic pole surface | ±1 mm | / |
| Rail-joint level | ±1 mm | The value under the condition of locked track temperature. |

Considering the deformation control requirements of the Maglev track and the influence of the tunnel crossing on the track on the bridge, the main influence of the tunnel crossing on the bridge under the working condition of this project is the differential settlement between the piers, which causes the differential deformation of the joint of the rail-row structure. The differential deformation of the rail-row joint has a direct impact on the track-height deviation and the rail-joint deviation. The photos of the rail expansion joint is shown in Figure 7.

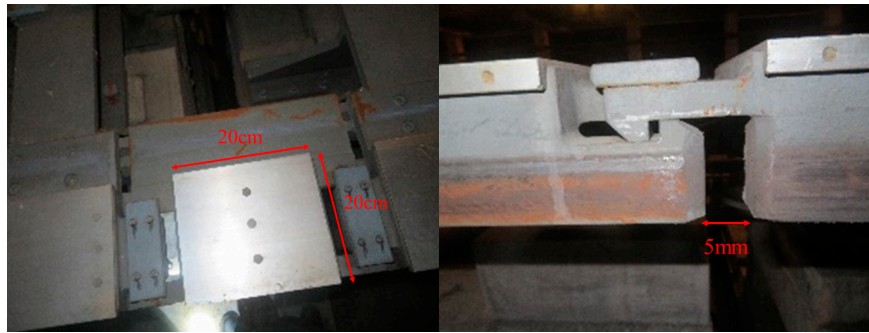

**Figure 7.** Rail expansion joint photos.

Considering the limit requirements of the track-height deviation and the rail-gap deviation, the differential settlement of the magnetic levitation track is the focus of this project. Considering the deformation difference between the bridge and the rail and the layout conditions of the on-site measuring points, differential deformation monitoring can be achieved by measuring the rail gap and the dislocation of the F-shaped steel. Finally, the project control indicators are determined as shown in Table 2.

**Table 2.** Deformation control index of the Maglev metro structure.

| Control Index | Warning Value (mm) | Alarming Value (mm) | Controlled Value (mm) |
|---|---|---|---|
| Pier structure settlement | 1.4 | 1.6 | 2.0 |
| Differential settlement between piers | 1.4 | 1.6 | 2.0 |
| Differential deformation between Maglev rail panels | – | – | +/−1.0 |
| The up and down and left and right deviation of F-shaped steel frame | – | – | +/−1.0 |

Note: warning value: starting value of the cautionary measures caused; alarming value: proposed starting value of structural deformation alarm; controlled value: allowable value of structural deformation.

## 3. Numerical Calculation Model

According to the geological conditions of the underpass section, a three-dimensional numerical calculation model was established using the finite element software Midas-GTS to analyze the deformation law of the existing Maglev viaduct pier and rail row. The overburden soil of the double-line tunnel is 16 m. According to the positional relationship between the old and new structures, the length is 200 m, the width is 100 m, and the thickness of the model is 50 m from the surface.

The D-P criterion was selected for soil constitution. Drucker and Prager proposed the generalized Mises yield and failure criterion considering the influence of hydrostatic pressure in 1952, abbreviated as the Drucker Prager (D-P) yield criterion. Its function can be expressed as:

$$F = \sqrt{J_2} - \alpha I_1 - \kappa = 0 \qquad (1)$$

in which $I_1$ = the first invariant of stress tensor; $J_2$ = the second invariant of deviator stress tensor; $\alpha$, $\kappa$ = material parameters.

Solid element simulation is used for soil and bridges to avoid stress concentration [31]. The initial support of the new tunnel is simulated by the shell element, the pile foundation is simulated by the beam element, the F rail and rail sleeper in the Maglev track are simulated by the beam element, and the connection is elastic. Grouting reinforcement is simulated by the solid element, and grouting reinforcement construction is simulated by endowing the model grouting reinforcement area with the original soil layer grouting properties. The ground surface (the upper surface of the model) is taken as the free boundary, and normal constraints are taken around and at the bottom of the model. The calculation model is shown in Figure 8 and the calculation parameters are shown in Table 3.

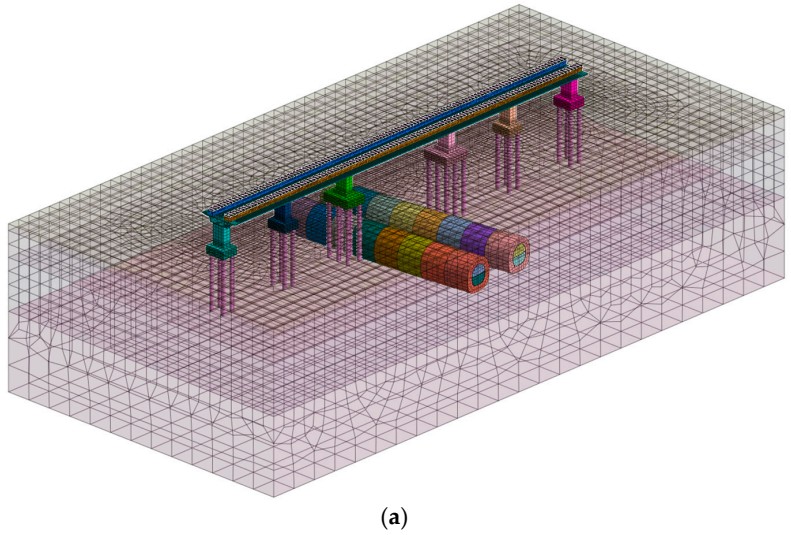

(**a**)

**Figure 8.** *Cont.*

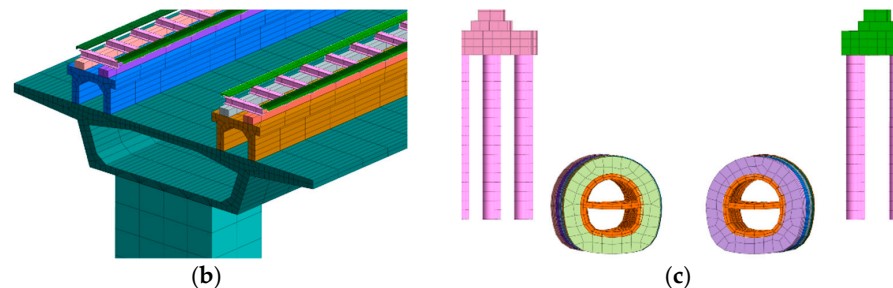

(**b**)  (**c**)

**Figure 8.** Numerical calculation model. (**a**) Overall model; (**b**) viaduct details; (**c**) cross-section details.

**Table 3.** Model parameters [32].

| Material Type | $\gamma$/(kN/m$^{-3}$) | E/MPa | C/kPa | $\varphi$/(°) | Poisson Ratio |
|---|---|---|---|---|---|
| Miscellaneous Fill | 17.0 | 12.0 | 10 | 5 | 0.30 |
| Pebble 5 | 21.0 | 140.0 | – | 35 | 0.25 |
| Pebble 7 | 21.5 | 200.0 | – | 40 | 0.25 |
| Pebble 9 | 21.5 | 300.0 | – | 42 | 0.22 |
| Grouting Area | 22.0 | 300.0 | – | 50 | 0.22 |
| Concrete C40 | 25.0 | $3.25 \times 10^4$ | – | – | 0.20 |
| Concrete C30 | 25.0 | $3.0 \times 10^4$ | – | – | 0.20 |
| Steel | 75.0 | $2.1 \times 10^5$ | – | – | 0.20 |

The first step of calculation is the derivation of the equilibrium of the initial ground stress, and then the excavation of subsurface tunnel is simulated. The excavation steps are properly simplified; the left and right lines are divided into eight sections, each about 8 m, and the right line is ahead of the left line by 16 m in the synchronous construction. The specific construction process is shown in Table 4 and Figure 9.

**Table 4.** Construction process.

| Steps No. | Specification |
|---|---|
| 1 | Right line step excavation 8 m, apply upper primary lining and middle partition plate. |
| 2 | ① Right line step excavation 8 m, apply upper primary lining and middle partition plate; ② Right line lower bench excavation 8 m, Apply lower primary lining. |
| 3 | ① Right line step excavation 8 m, apply upper primary lining and middle partition plate; ② Right line lower bench excavation 8 m, apply lower primary lining; ③ Left line step excavation 8 m, apply upper primary lining and middle partition plate. |
| 4~10 | Left and right line synchronous excavation according to the upper and lower steps. |
| 11 | Complete all middle partition middle partition removal, complete all secondary lining. |

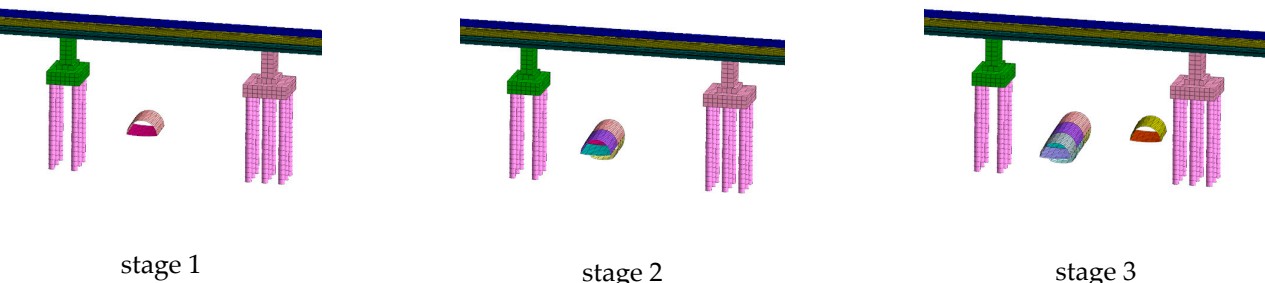

stage 1  stage 2  stage 3

**Figure 9.** *Cont*.

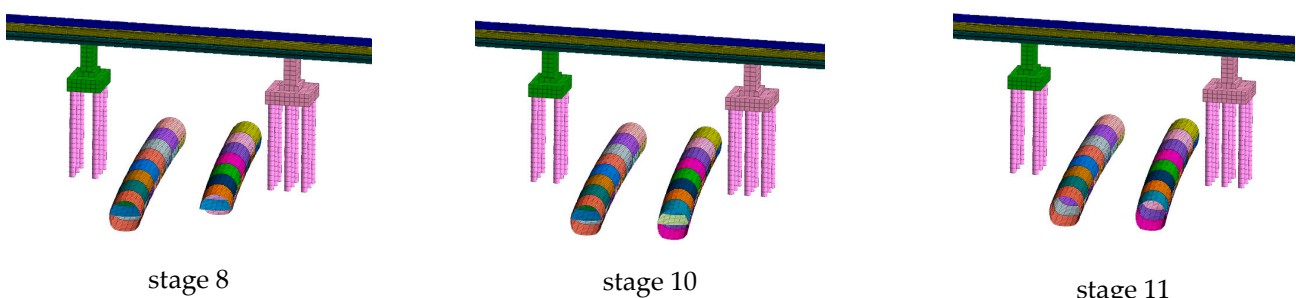

stage 8        stage 10        stage 11

**Figure 9.** Construction process simulation.

## 4. Analysis of Calculation Results

Predictive Analysis of Settlement Deformation of Magnetically Suspended Viaduct Pier and Rail under Deformation Control Scheme.

### 4.1. Deformation Analysis of Existing Bridge Piers

In each pier on the two left and right observation points, close to the tunnel excavation side and away from the tunnel excavation measurement, respectively, the construction stage pier observation points in each process of the vertical change time curve are drawn, as shown in Figure 10.

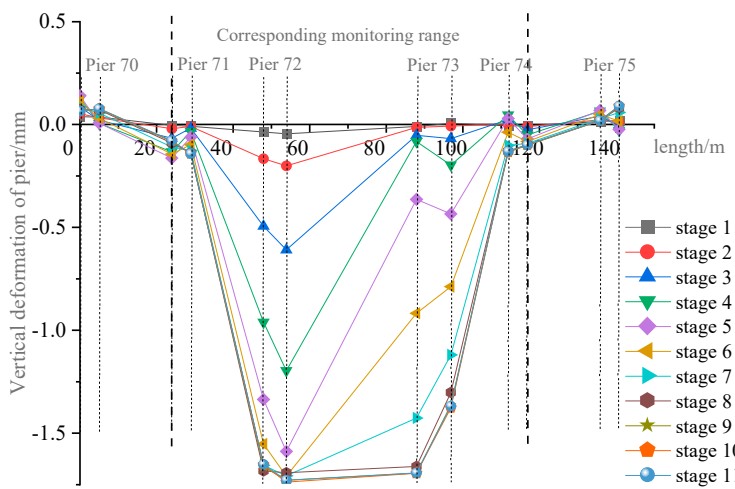

**Figure 10.** Vertical deformation curve of the existing pier.

On the whole, the trend of deformation of piers 71#~74# in each construction stage is caused by settlement. The deformation law is analyzed. As the construction progresses, the deformation of each pier gradually increases. For construction stages one, two, ten, and eleven, the influence of tunnel excavation on the existing structure is small, and the deformation value of the existing pier changes little. The main deformation of each pier occurs from process three to process seven. The maximum deformation value occurs in piers 72# and 73#, that is, two piers adjacent to the new tunnel, which has no obvious effect on other piers; among them, the maximum deformation value of pier 72# is −1.73 mm and the maximum deformation value of pier 73# is −1.70 mm, with both occurring in process 11.

In order to more intuitively see the change in the maximum deformation value of each pier in each construction process, the maximum deformation value of each pier in each process is extracted to draw the vertical deformation time history curve, as shown in Figure 11. The deformation of piers 71#, 72#, 73#, and 74# is analyzed with monitoring points.

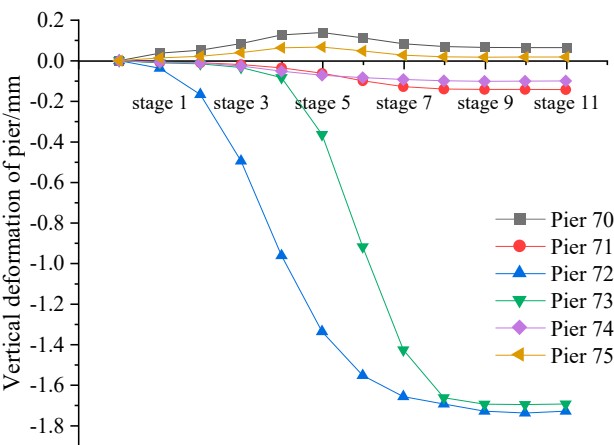

**Figure 11.** Vertical deformation time history diagram of the existing pier.

It can be seen that piers 72# and 73# adjacent to the new tunnel are most affected by the entire tunnel excavation. Because the left line of the new tunnel is excavated two processes later than the right line, the deformation of pier 74# has a certain lag compared to pier 73#. The fourth process is the excavation of the right line tunnel to the point of the underpass, so the right line tunnel excavation in the third to fifth stages of the process has the greatest impact on the deformation of pier 72#, and the deformation value accounts for 68.9% of the total. Process 6 is the excavation of the left line tunnel to the point of the underpass; from process 5 to process 7, the right line tunnel excavation has the greatest influence on the deformation of pier 74#, accounting for 77.8% of the total deformation value. In the sixth process, the settlement of pier 72# tends to be stable after the excavation face of the right line tunnel passes through the point of the underpass and advances forward about 16 m. Process eight is the excavation of the left line tunnel face through the point of the underpass and advances forward about 16 m. The differential settlement between piers occurs between 72# and 71#, and 73# and 74#, and the differential settlement is 1.73 mm and 1.70 mm, respectively.

### 4.2. Track Structure Deformation Analysis

The influence of the excavation of the new tunnel on the magnetic levitation viaduct is finally reflected in the track structure. Therefore, the deformation values of the left and right magnetic levitation rails in each process are extracted and the dotted line diagram is drawn, as shown in Figures 12 and 13.

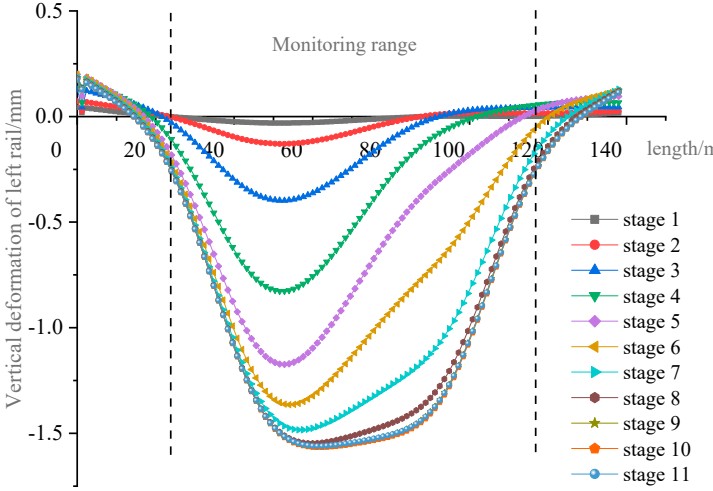

**Figure 12.** Settlement curve of the existing left rail.

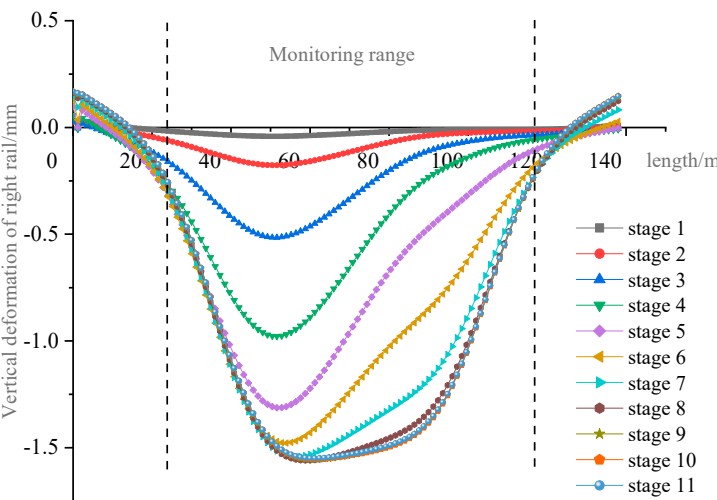

**Figure 13.** Settlement curve of the existing right rail.

The upper rails corresponding to piers 71#, 72#, 73#, and 74# and the middle rails of the bridge are named according to the monitoring points. Five rail-row measuring points are arranged along the bridge, one for each of the corresponding four pier positions and one for crossing the mid-span. The layout of monitoring points is shown in Figure 14.

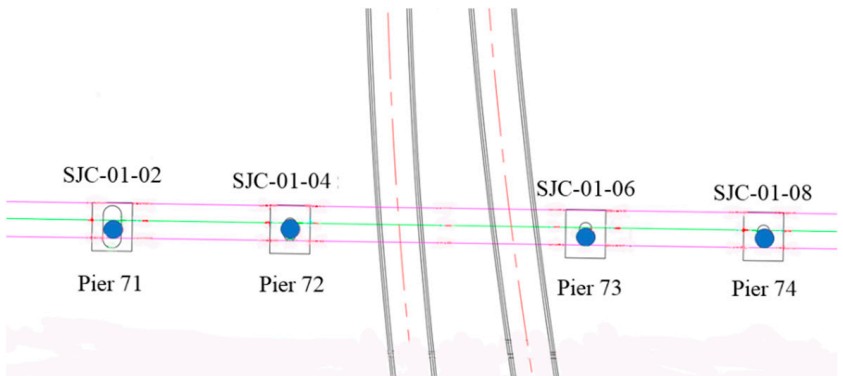

**Figure 14.** Layout of monitoring points.

From the diagram, it can be seen that the settlement trend of the rail row is 'U'-shaped, and the deformation law of the rail row on the left and right lines is almost the same as the deformation trend of the corresponding pier. The deformation values of the left and right line rails at the end of process 11 were analyzed and the deformation diagram was drawn, as shown in Figure 15. It can be seen that the maximum settlement value of each point of the left rail is slightly larger than that of the right rail, so the settlement value of the left rail is taken as an example for analysis. The maximum settlement of each point occurs in process 11. SJC-01-04 is the maximum deformation point and the maximum cumulative deformation point, and the maximum settlement value is 1.56 mm. The maximum settlement value of SJC-01-05 is 1.54 mm, and the maximum settlement value of SJC-01-06 is 1.52 mm. Existing rail deformation values meet the control index requirements.

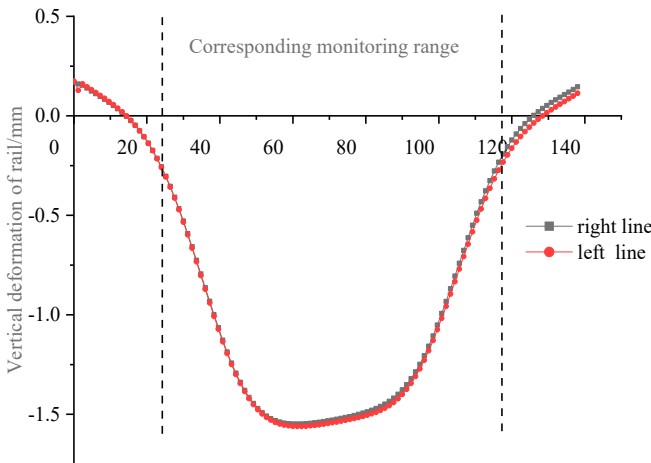

**Figure 15.** Comparison of maximum deformation of existing left and right rails.

In order to more intuitively show the deformation trend of the rail row in the whole construction process, the vertical deformation values of the typical points SJC-01-02, SJC-01-04, SJC-01-05, SJC-01-06, and SJC-01-08 on the rail row in each process were extracted, and the deformation time history curve was drawn, as shown in Figure 16.

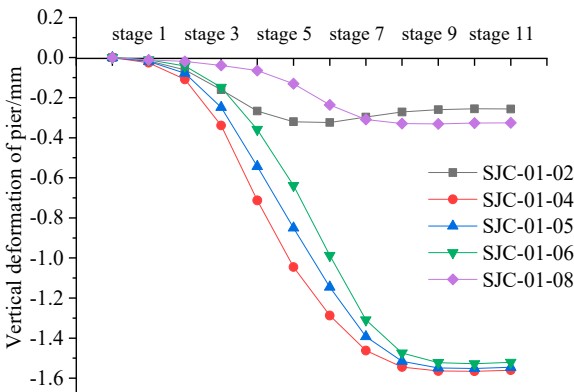

**Figure 16.** Time history curve of existing left rail deformation.

From the diagram, it can be seen that the vertical deformation of each typical point is the settlement trend. In the initial stage of construction, the excavation section of the new tunnel is far away from the existing structure, so the deformation value is small. As the excavation advances, the deformation value and deformation rate also increase. The deformation rate reaches the maximum when the excavation section reaches the undercrossing point, and then the deformation rate decreases continuously until the deformation reaches the extreme value when the excavation of the process 11 tunnel is completed. Because the right line tunnel adjacent to pier 72# is excavated first and the left line tunnel adjacent to pier 73# is excavated later, the settlement time of pier 73# is later than that of pier 72#. There is a maximum differential settlement of 0.5 mm between piers 72# and 73# in the construction process, but the final differential settlement tends to 0. The final cumulative calculated settlement of the pier is 1.73 mm and the maximum cumulative calculated settlement of the Maglev rail is 1.56 mm. From the relationship between the settlement of the pier and the rail row, it can also be seen that the settlement deformation of the pier will be attenuated after the pier, bearing, bridge, and track beam are finally transmitted to the rail row. Because the crossing section is a continuous beam, the deformation curve of the rail is relatively flat, and there is no obvious differential settlement between the rails.

## 5. Field Measurement Analysis

In order to further verify the reliability of the calculation results of the numerical model, the field monitoring data of the vertical deformation of the existing magnetic levitation rail are analyzed.

### 5.1. Existing Rail Deformation Measurement Analysis

Considering that the vertical deformation of the left and right rails of the magnetic levitation viaduct is not significantly different, the vertical deformation monitoring data of the left rails are mainly analyzed. The vertical deformation monitoring points SJC-01-02, SJC-01-04, SJC-01-06, and SJC-01-08 of the upper left rail of piers 71#, 72#, 73#, and 74# and the vertical deformation monitoring point SJC-01-05 of the left rail of the viaduct are taken as typical points to draw the deformation time diagram, as shown in Figure 17.

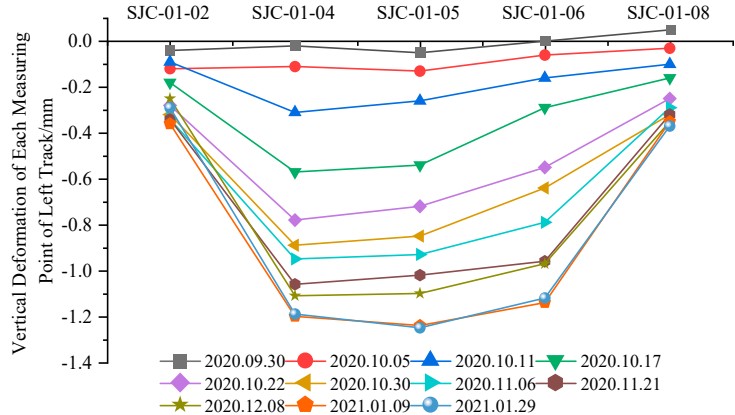

**Figure 17.** Monitoring values of the settlement curve of the existing left track.

On the whole, the rails at each monitoring point show a settlement trend with the advancement of the construction stage, and the settlement trend is 'U'-shaped. The measured deformation trend is consistent with the numerical simulation deformation trend. The maximum cumulative vertical deformation occurs at the measuring point SJC-01-04~06, that is, the middle span of the viaduct; the maximum cumulative vertical deformation value of SJC-01-04 (left rail track on the upper part of pier 72#) is −1.19 mm; the maximum cumulative vertical deformation of SJC-01-06 (the left rail at the top of pier 73#) is −1.12 mm. The maximum cumulative deformation of SJC-01-02 and SJC-01-08 is only 0.29 mm and 0.37 mm, respectively.

The time history deformation data of the vertical deformation monitoring points SJC-01-02, SJC-01-04, SJC-01-05, SJC-01-06, and SJC-01-08 of the rail row are analyzed. As shown in Figure 18, in the process of advancing the tunnel excavation face, the settlement deformation of the measuring point SJC-01-04 is first larger than that of other measuring points. With the excavation of the tunnel, and under the superposition of the excavation of the right line and the left line tunnel, the settlement deformation gap between the measuring points SJC-01-06 and SJC-01-04 gradually narrows. As the excavation face of the left line tunnel on the right line passes through the undercrossing point, the settlement rate of each monitoring point successively begins to decrease, and the settlement deformation gradually tends to be stable. The settlement law of SJC-01-02 and SJC-01-08 measuring points is similar to that of the above measuring points, but the settlement value is small.

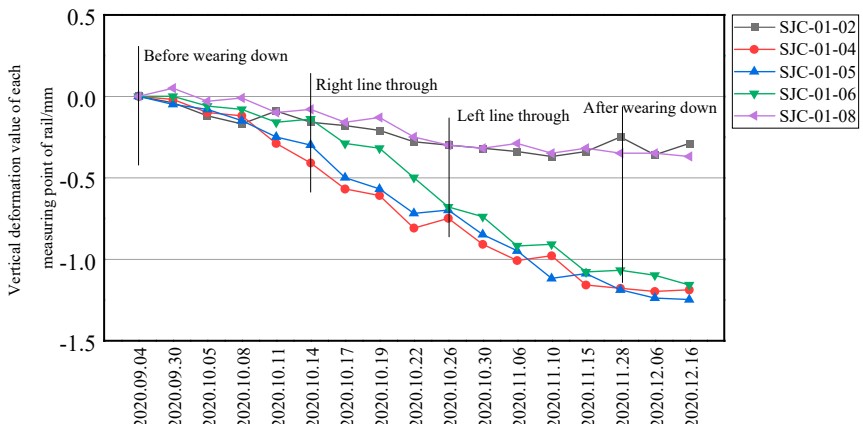

**Figure 18.** The monitoring values of the existing left track settlement time history curve.

*5.2. Comparison of Existing Rail Deformation Results*

In order to more intuitively see the consistency between the monitoring results and the numerical simulation results, the simulated values of the final deformation of the existing magnetic levitation rails in the monitoring range are compared with the measured values, as shown in Figures 19 and 20.

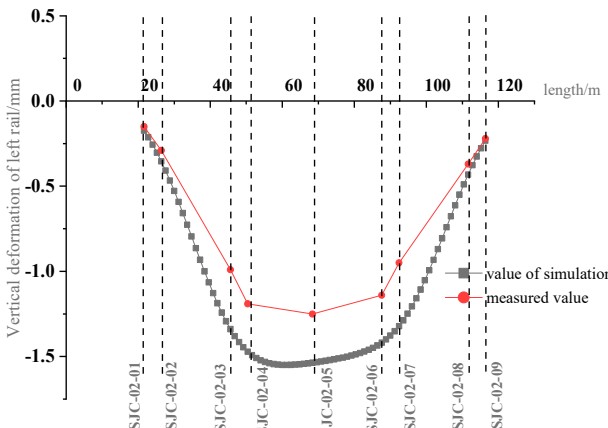

**Figure 19.** Comparison between simulation values and monitoring values of the deformation of the existing left track.

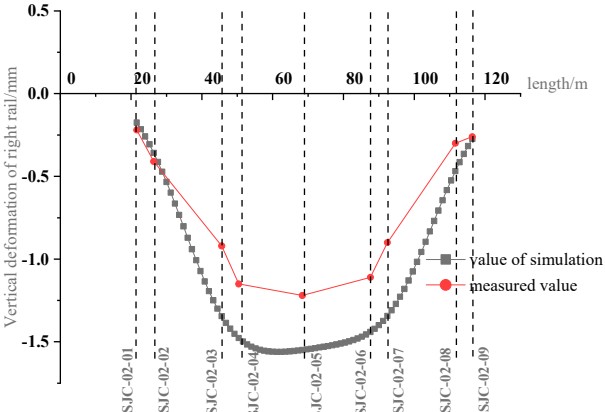

**Figure 20.** Comparison between simulation values and monitoring values of the deformation of the existing right track.

From the diagram, it can be seen that the maximum vertical deformation value of the existing left line rail row measured in the field is −1.19 mm and the maximum vertical deformation value obtained by numerical simulation is −1.56 mm; the measured value of the maximum vertical deformation of the right rail is −1.22 mm and the simulated value is −1.52 mm. The deformation simulation value is slightly larger than the field-measured value. The numerical simulation results of the existing magnetic levitation rails are basically consistent with the final vertical deformation trend of the field-measured data. Therefore, it can be considered that both numerical simulation and field monitoring can better reflect the deformation trend of the existing structure.

*5.3. Comparison of Differential Deformation Results of Existing Rails*

The on-site rail gap was examined using the vernier caliper method, with a Leica laser tracker, to measure the surface deviation of the F-shaped steel. The measuring principle is shown in Figures 21 and 22.

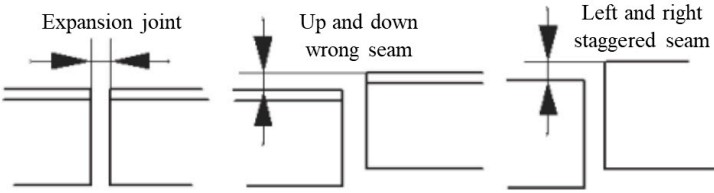

**Figure 21.** Track seam monitoring method.

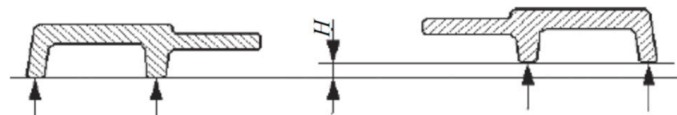

**Figure 22.** F-shaped steel deviation monitoring method (H: differential height).

According to the measurement results, there is no obvious relative change in the surface deviation between the rail joint and the F-shaped steel during the tunnel crossing process, and the smoothness of the magnetic levitation track is not significantly affected, indicating that the whole settlement curve is relatively smooth. The settlement of the bridge, of 1.22 mm, has little effect on the magnetic levitation track, and the operation of the magnetic levitation track is safe.

## 6. Conclusions

A three-dimensional stratum–structure model was established for the two-line subway tunnel excavation under the existing Beijing S1 line medium-low-speed Maglev track project. Combined with the measured results of the project site, the following conclusions can be stated:

(1) According to the particularity of the Maglev track structure, the main deformation control point of this project is the differential deformation of the magnetic levitation track. Considering the layout conditions of the field measuring points, the deformation difference between the Maglev rail panels can be monitored by measuring the rail joint and the dislocation of the F-shaped steel. The actual monitoring process data prove that the method is feasible.

(2) The maximum deformation value of the pier adjacent to the tunnel crossing position is −1.73 mm and the maximum vertical deformation value of the magnetic levitation rail is −1.56 mm. It can be seen that the settlement deformation of the pier will be attenuated after passing through the pier, bearing, bridge, and track beam to the rail. Because the bridge is a continuous beam, the deformation curve of the rail is relatively flat, and there is no obvious differential settlement between the rails.

(3) The measured maximum vertical deformation of the Maglev rail is −1.25 mm, which is slightly smaller than the numerical calculation. Due to the slow deformation curve, no obvious dislocation or differential deformation were found in the field rail joint and F-shaped steel during the monitoring process. The numerical simulation results of the existing magnetic levitation rail were basically consistent with the final vertical deformation trend of the field-measured data. Numerical simulation prediction combined with field measurement can better reflect the deformation trend of the existing structure.

(4) In the project of the double-line tunnel crossing under the existing Maglev track viaduct, the deformation of the existing structure can be effectively controlled by using deep-hole grouting to reinforce the surrounding soil of the tunnel in advance. During the crossing process, jack lifting is used on the upper structure of the bridge as an emergency measure. This study can provide a reference for similar projects.

**Author Contributions:** Conceptualization, H.P.; methodology, H.P. and Z.W.; software, Z.W. and X.P.; validation, X.X. and Z.L.; investigation, H.P. and X.X.; resources, H.P.; data curation, Z.W. and X.P.; writing—original draft preparation, X.X.; writing—review and editing, Z.W. and X.P.; visualization, H.P.; project administration, H.P. and Z.L.; funding acquisition, H.P. All authors have read and agreed to the published version of the manuscript.

**Funding:** This research was funded by [Science and Technology Research and Development Program Project of China National Railway Group Co.] grant number [N2020G009].

**Institutional Review Board Statement:** Not applicable.

**Informed Consent Statement:** Not applicable.

**Data Availability Statement:** Not applicable.

**Conflicts of Interest:** The authors declare no conflict of interest.

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
