# Peer review of "Deformation Control of the Existing Medium-Low-Speed Maglev Metro Viaduct over a Double-Line Bored Tunnel"

_applsci, doi:10.3390/app13116659_

Round 1
Reviewer 1 Report
There are some points that should be addressed before making a decision on this manuscript:
1. The literature review of the manuscript should be revisited; particularly, the analytical treatment of soil-structure interaction has been ignored. Authors are referred to
a. Journal of Engineering Mechanics 140.7 (2014): 04014048.
b. Soil Dynamics and Earthquake Engineering 63 (2014): 184-192.
c. Civil Engineering Infrastructures Journal 47.1 (2014): 125-138.
d. International Journal of Solids and Structures 90 (2016): 251-260.
e. Computers and Geotechnics 119 (2020): 103347.
f. Applied Mathematical Modelling 77 (2020): 663-689.
2. Please add scale bars to the images (Figs. 6 and 7 for instance) used in the manuscript.
3. Not sure if ‘process’ would be a correct word choice in Fig. 9! Also, please make sure to define all terms and variable used in the manuscript clearly.
4. Please shorten the number of figures, 21 figures is too much in a 15-page manuscript.
Author Response
Point 1: The literature review of the manuscript should be revisited; particularly, the analytical treatment of soil-structure interaction has been ignored. Authors are referred to
a Journal of Engineering Mechanics 140.7 (2014): 04014048.
b Soil Dynamics and Earthquake Engineering 63 (2014): 184-192.
c Civil Engineering Infrastructures Journal 47.1 (2014): 125-138.
d International Journal of Solids and Structures 90 (2016): 251-260.
e Computers and Geotechnics 119 (2020): 103347.
f Applied Mathematical Modelling 77 (2020): 663-689.
Response 1: Sorry for the missing content. We have carefully examined the literature review and added the content of soil-structure interaction. Which is presented in the third paragraph of section 1 as following.
“Meanwhile, the soil-structure interaction is also a key point in scholars’ research. Tullini, N et al. [18] studied the buckling deformation of finite length Timoshenko beams in frictionless contact with an elastic substrate in a generalized plane stress or plane strain regime. Gao, Q et al. [19] studied the soil-structure interaction emanating from seismic stationary random excitations by using the pseudo-excitation method in com-bination with the precise integration method, the result shows that soil–structure interaction is very significant under the condition of large structure mass or soft soil.”
Point 2: Please add scale bars to the images (Figs. 6 and 7 for instance) used in the manuscript.
Response 2: We have added scale bars to Figs. 6 and 7.
Point 3: Not sure if ‘process’ would be a correct word choice in Fig. 9! Also, please make sure to define all terms and variable used in the manuscript clearly.
Response 3: We have double-checked the terminology in the article and changed "process" to "stage" in Figures 10, 11, 12, 13, and 16.
Point 4: Please shorten the number of figures, 21 figures is too much in a 15-page manuscript.
Response 4: Thanks so much for this comment. We have shorten the number of figures, to optimize the overall structure of the manuscript.

Reviewer 2 Report
In this work, the authors analyze the deformation that occurred in the bridge piers and track panels of an existing viaduct while constructing of a double line subway tunnel beneath it. They also compared their analysis with monitoring data taken in the field, which showed good agreement with the simulations.
After thoroughly reviewing this manuscript, I cannot recommend publishing this work in the journal at this stage for the following reasons:
1) It is not clear how the authors carry out the simulations, which is a crucial aspect of this study. The authors should provide more details on the constructions processes (it is not very clear to me the processes) and how they modeled them in the simulations (there is not details regarding this in the manuscript).
Minor things:
2) The font size in the figures should be increased, as it is difficult to read them in the current size.
3) The author should also indicate the name of the parts that they refer to in the figures to improve the clarity.
4)The captions of the figures need improvement, as they are sometimes too plain and lack meaningful information.
Author Response
Point 1: It is not clear how the authors carry out the simulations, which is a crucial aspect of this study. The authors should provide more details on the constructions processes (it is not very clear to me the processes) and how they modeled them in the simulations (there is not details regarding this in the manuscript).
Response 1: The construction process have been supplemented in detail in the text, as shown in Figures 9.
Point 2: The font size in the figures should be increased, as it is difficult to read them in the current size.
Response 2: We modified the figure sizes for Figures 1-4, 19, 20, and 21 in the article.
Point 3: The author should also indicate the name of the parts that they refer to in the figures to improve the clarity.
Response 3: We modified the article to improve the clarity of Figures 1-5.
Point 4: The captions of the figures need improvement, as they are sometimes too plain and lack meaningful information.
Response 4: We adjusted the figure for Figures 1-5, 19, 20, and 21 in the article.

Reviewer 3 Report
My recommendation is to follow these comments, to consider the manuscript for possible publication:
-What is the novelty of the present study?
-Several mistakes along the paper: Lines 15, 68, 73, 193, 196 etc.
-Adjust size of numbers and words for figures 1-5, and 20.
-maglev or Maglev? Please unify terms in the paper.
-Explain D-P criterion in detail.
-How were the parameters in Table 3 obtained?
-Section 3 - computation module - needs to be improved.
-The figure captions are different, please unify.
-Does figure 21 include text or numbers?
-The phrase "...the study can provide reference for similar projects" is repeated in the abstract, introduction and conclusions. Please improve this aspect.
-It would be interesting to compare the results obtained with what is observed in these systems in other countries. The references used are very specific.
Author Response
Point 1: What is the novelty of the present study?.
Response 1: The novelty of the study is mainly about the special requirements of track system of Maglev Metro, which is different from normal ballastless track. Therefore this project carried out several measures to make sure the safety of the Maglev Metro.
Point 2: Several mistakes along the paper: Lines 15, 68, 73, 193, 196 etc.
Response 2: Thanks so much for pointing out these small details. The manuscript has been revisited and punctuation errors, font formatting errors etc. have been corrected.
Point 3: Adjust size of numbers and words for figures 1-5, and 20.
Response 3: We have increased the size of the figures and text for Figures 1-5 and 21 in the article.
Point 4: maglev or Maglev? Please unify terms in the paper.
Response 4: We have revised the entire text to "Maglev".
Point 5: Explain D-P criterion in detail.
Response 5: Sorry for the missing content. We have added the content of D-P criterion explanation. Which is presented in the second paragraph of section 3 as following.
Drucker and Prager proposed the generalized Mises yield and failure criterion consid-ering the influence of hydrostatic pressure in 1952, abbreviated as Drucker Prager (D-P) yield criterion, its function can be expressed as:
in which I1 =the first invariant of stress tensor; J2 =the second invariant of deviator stress tensor; α, κ =material parameters.
Point 6: How were the parameters in Table 3 obtained?
Response 6: They are obtained from the literature review 32 in this manuscript.
Point 7: Section 3 - computation module - needs to be improved.
Response 7:The name of section 3 change to Numerical calculation model, and the contents are revised.
Point 8: The figure captions are different, please unify.
Response 8: Thanks a lot for this comment. Format and case errors have been corrected in Fig.1 2 8 17 19 20 21.
Point 9: Does figure 21 include text or numbers?
Response 9:Yes, H means Differential Height, which is added in the article.
Point 10: The phrase "...the study can provide reference for similar projects" is repeated in the abstract, introduction and conclusions. Please improve this aspect.
Response 10:The one in introduction is deleted.
Point 11: It would be interesting to compare the results obtained with what is observed in these systems in other countries. The references used are very specific.
Response 11: Sorry,we haven’t found much similar case in China and other countries, as the Meglev track are not widely used. Therefore this article is mainly refer to this specific project. And show how to take measures in advance.

Round 2
Reviewer 2 Report
The authors have addressed all my questions and concerns and modified accordingly the manuscript.
Reviewer 3 Report
I recommend its publication in the current state.